# The breadth of the Mexican Transition Zone as defined by its flowering plant generic flora

**Villaseñor José Luis**[ID]<sup></sup>\*, **Enrique Ortiz**<sup></sup>, **Claudio Delgadillo-Moya**<sup></sup>, **Diego Juárez**<sup></sup>

Instituto de Biología, Universidad Nacional Autónoma de México, Mexico City, Mexico

☯ These authors contributed equally to this work.
\* vrios@ib.unam.mx

**Data Availability Statement:** All relevant data are within the manuscript and its Supporting Information files.

**Funding:** Work supported by the Universidad Nacional Autónoma de México - Programa de

## Abstract

Biogeographic regions are defined by taxa with similar distribution patterns. Flowering plants have been widely used to propose biogeographic regionalization schemes because of shared historical or ecological factors that determine their distribution. The Mexican Transition Zone represents the boundary between the Nearctic and Neotropical kingdoms; however, there is no general agreement about the limits and extent of this region. Despite the significance of its role in the history of Mexican biota, no study involving a set of relevant plant taxa validates the magnitude of the Mexican Transition Zone. This work attempts to determine the proportion of flowering plant families and genera that characterize the biogeographic kingdoms and the Mexican Transition Zone. Through identification of distinctive genera it is shown that the Mexican Transition Zone includes the mountains of Mexico, from Oaxaca northwards. The results provide a broad view of the distribution patterns of the flora of Mexico and allow the evaluation of relationships and floristic affinities.

## Introduction

The boundary between the Nearctic and Neotropical floristic realms is found in Mexico [1, 2]. Geologically, much of the tropical area currently occupied by Mexico and Central America belongs to southwestern Laurasia, while Neotropical South America belongs to Gondwana [3]. These distinct geological histories must significantly define the composition of the Mexican flora since North America (Laurasia) was isolated from South America (Gondwana) until the Pliocene, when the closure of the Isthmus of Panama occurred.

Although it has been argued that the existence of oceanic bridges and island hoping allowed dispersal between both American sub-continents since the end of the Cretaceous [4], it is evident that the main elements in the flora of Mexico (Nearctic and Neotropical) constitute the chief cenocrons (sets of taxa with a common biogeographic history, *sensu* Morrone [5]) that explain its regional history. Consequently, those taxa distributed in North America and Mexico and those shared between Mexico and South America may help to identify the role of the kingdoms in the composition of the Mexican flora and to determine their boundaries.

Plants, especially flowering plants, have been used to propose biogeographic regionalization schemes (provincialism *sensu* Brown and Gibson [6]). The first scheme was proposed by de

Apoyo a Proyectos de Investigación e Innovación Tecnológica (PAPIIT IN209519) to JLV. There was no additional external funding received for this study.

**Competing interests:** The authors have declared that no competing interests exist.

Candolle; he divided the world into 20 areas of endemism by using mainly elements of the Asteraceae family, identifying Mexico as one such area of endemism [2, 7]. Takhtajan [1] proposed one of the most important regionalization outlines based on plants ("floristic system"), detailing specific criteria for the definition of its 'phytochoria' (or natural floristic areas). In his hierarchical system, Takhtajan states that kingdoms are characterized by both families and endemic genera. Regions within kingdoms are also characterized by the presence of endemic genera and by high percentages of endemic species. Although quantities are not clearly defined, the presence of endemic elements is an indispensable requirement to identify kingdoms and regions. The territory of Mexico, for example, includes primarily North American (Nearctic element) and South American (Neotropical element) as well as endemic families (Table 1 and Fig 1). These families justify recognition of the two kingdoms in its territory.

According to Villaseñor [8], the Mexican flora contains about 2706 genera of flowering plants. He documented the worldwide distribution of 2703 of them as indicated in Table 2 which can be divided into four main floristic elements (*i.e.*, groups of taxa that have similar geographical distribution, *sensu* Birks [9]). For the purposes of this study, Nearctic affinity elements (or North American *sensu* Cox [2]) are defined as the genera endemic to Mexico and those distributed from North America to Mexico and Central America. Furthermore, the Neotropical affinity elements (or South American *sensu* Cox [2]) comprise genera distributed from South America to Mexico. A third element (Pan-American) includes genera with wider distribution throughout the continent. The last element includes the genera that are also known in the Old World and may be found in all or nearly all continents (Holarctic and Paleo-tropical), *i.e.*, some are widespread distributed

## The Mexican Transition Zone

Biogeographic regions have an identity that is revealed by taxa with similar distribution patterns, responding to either historical or ecological factors. Regionalization schemes have been published globally [1, 2, 10] or nationally [11, 12, 13, 14]. Morrone [14] introduced an exhaustive review of different regionalization proposals for Mexico.

A consensus among biogeographical proposals maintains that Mexico is in the transition between the Nearctic and Neotropical biogeographic kingdoms. However, there is no general agreement in the number of regions or provinces in each kingdom, perhaps due to the use of different taxa with different biogeographic histories or the analytical strategies to define them.

**Table 1. Nearctic and Neotropical families that find their distribution limit in Mexico.** The endemic families of Mexico are indicated by an asterisk.

| Nearctic Kingdom | Neotropical Kingdom |
|---|---|
| Crossosomataceae | Achatocarpaceae |
| Fouquieriaceae | Alstroemeriaceae |
| Guamatelaceae | Brunelliaceae |
| Iteaceae* | Cyclanthaceae |
| Petenaeaceae | Lacistemataceae |
| Plocospermataceae | Marcgraviaceae |
| Simmondsiaceae | Muntingiaceae |
| Setchellanthaceae* | Phyllonomaceae |
| Stegnospermataceae | Picramniaceae |
| Ticodendraceae | Schlegeliaceae |
| | Tovariaceae |

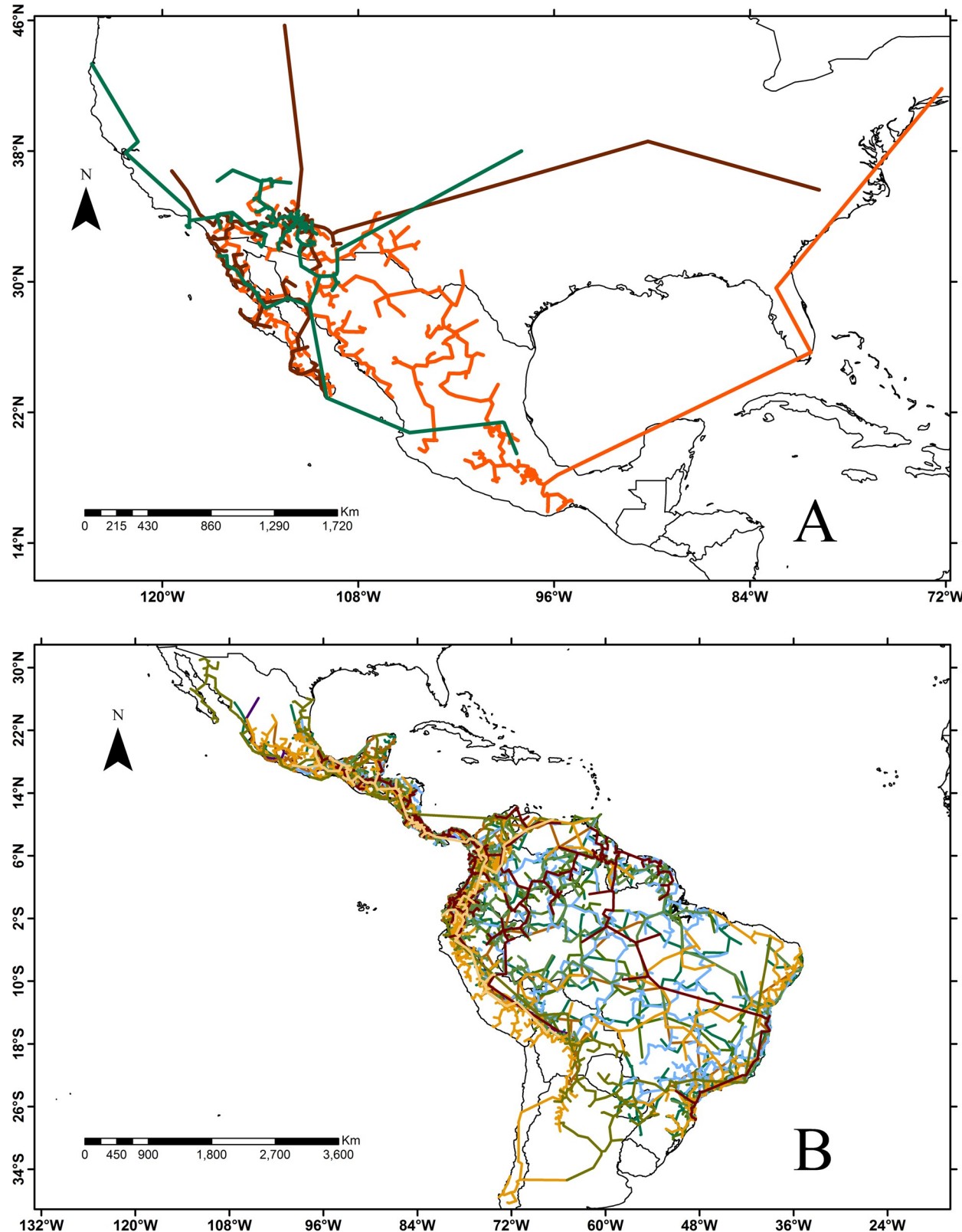

**Fig 1.** Examples of distribution (biogeographic tracks) of families restricted to the Nearctic (A) and Neotropical (B) Kingdoms whose distribution limit is in Mexico. (A) Representative families: Crossosomataceae, Fouquieriaceae, Simmondsiaceae; (B) Representative families: Achatocarpaceae, Alstroemeriaceae, Cyclanthaceae, Lacistemataceae, Marcgraviaceae, Phyllonomaceae, Picramniaceae, Schlegeliaceae, Tovariaceae.

**Table 2. Geographical distribution of the genera of flowering plants of Mexico.** Distinctive genera are those with half or more of their species in the country. Number of genera endemic to Mexico is given in parentheses.

| Distribution | Floristic Element | Total Genera | Distinctive genera |
|---|---|---|---|
| North America to Mexico | Nearctic | 600 (232) | 482 |
| North to Central America | Nearctic | 35 | 30 |
| Mexico to Central America | Nearctic | 69 | 62 |
| Mexico to South America | Neotropical | 772 | 282 |
| Broad in the Americas | Pan-American | 323 | 149 |
| Reaching the Old World | Widespread | 904 | 147 |
| | | **2703** | **1152** |

For example, CONABIO [15] divides Mexico into 19 biogeographic provinces (Fig 2), while Rzedowski [11] recognizes only 17, and Morrone [14] only 14.

The geographical location of provinces within kingdoms is likewise heterogeneous. For example, Rzedowski [11] places the provinces Baja California, Mexican Highland, and Pacific Northwest Coast as part of the Mexican Xerophytic region within the Neotropical Kingdom. In contrast, Morrone [14] places them as part of the Nearctic Region (zoological equivalent to the Nearctic Kingdom).

Differences in placement criteria of the biogeographic provinces make the definition of limits between kingdoms diffuse and poorly understood. Traditionally, such limits have been described as lines on maps that do not really reflect the existing biological complexity there, as they are overlap areas of characteristic members of each kingdom. The areas where overlapping elements of two kingdoms or biogeographic regions coincide are known as transition zones. Ferro and Morrone [16] review the concept of transition zone and discuss ways to detect and define its extent. Likewise, Morrone [5] discusses articles aimed at defining and geographically locating the Mexican Transition Zone (MTZ onwards), i.e., the limiting area between the Nearctic and Neotropical Kingdoms or Regions.

The MTZ has been more widely studied by zoologists than by botanists [5, 17, 18, 19] without conclusively defining a region. Its breadth may range from the southwestern United States to Central America or restrict it to the main mountain ranges of Mexico, from Chiapas to northern Mexico. Rzedowski [11] was the first botanist to discuss the existence of a region between the Holarctic and Neotropical Kingdoms where their elements intermingle without a clear dominance of either. He did not call it a transition zone but placed it as intermediate between both kingdoms, comprising the Mesoamerican Mountain Region, which also includes all the mountain ranges of the country, from Chiapas northwards. Among the few studies of the MTZ with plants are those of Delgadillo [20] who discusses the transitional role of Mexico in the distribution of mosses, and Contreras-Medina et al. [21] with gymnosperms, who considered the MTZ from the southern United States to the contiguous part of Central America with Mexico.

Despite their role in the history of the Mexican biota, there is still no study to test the magnitude of MTZ using a relevant set of taxa from the flora of Mexico. Therefore, this work aims to determine the breadth of the MTZ based on the distribution of the genera of flowering plants occurring in Mexico. A second goal is to evaluate the coincidence of the proposed biogeographical regions (CONABIO [15]) with the distribution of distinctive genera of the flora of Mexico inside and outside of the suggested MTZ.

To attain these objectives, the known distribution of the genera of the flora of Mexico was used to determine the proportion of genera that characterize the biogeographic kingdoms and the MTZ, and how well they characterize the biogeographical regions proposed by CONABIO [15].

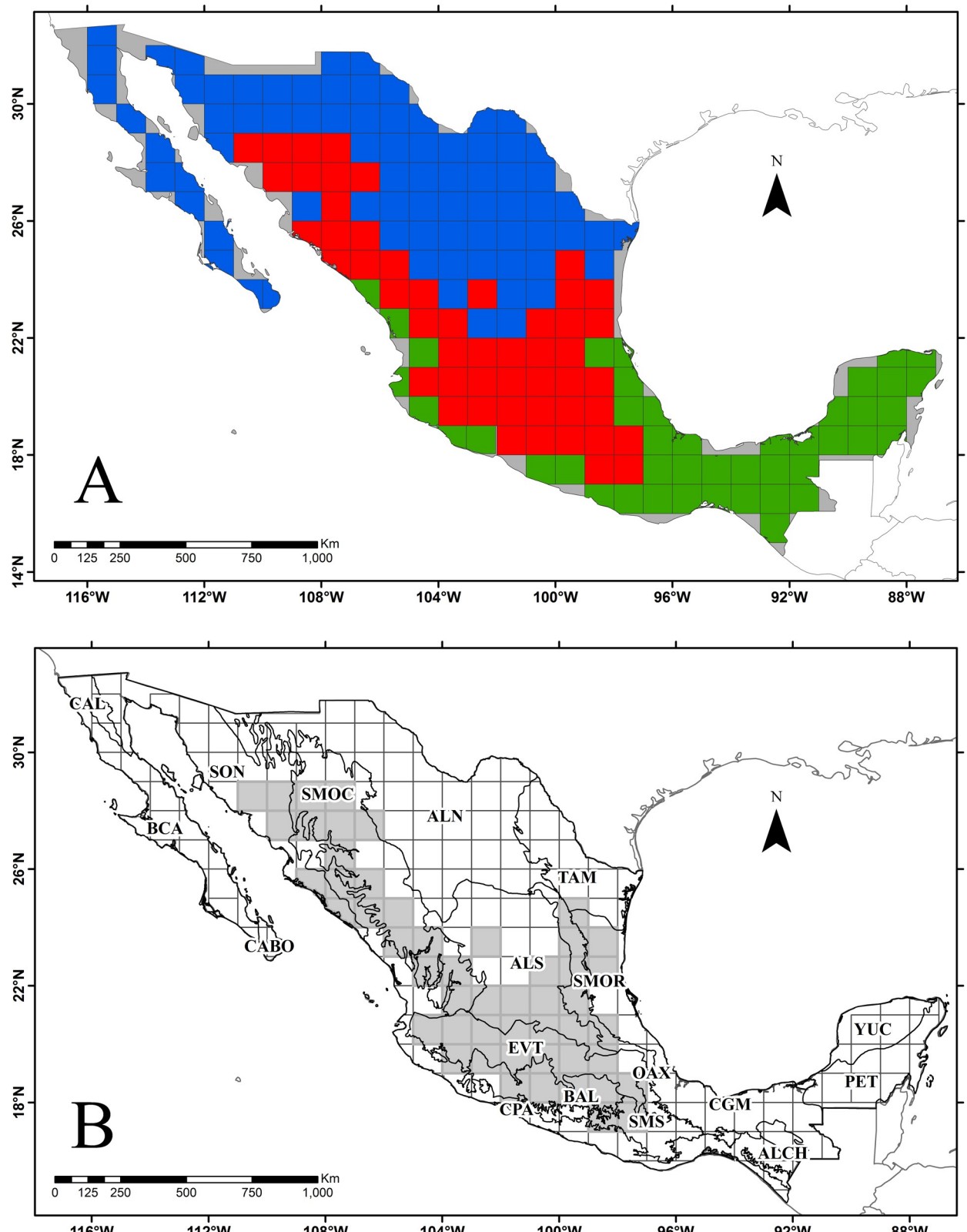

**Fig 2.** A) Assignment of the 175 grid squares in which Mexico was divided according to its geographical affinity. Blue: Nearctic Kingdom, Green: Neotropical Kingdom, Red: Mexican Transition Zone. The gray zone indicates the peripheral grid squares with less than half the total area that were merged with their neighboring squares. B) Mexican biogeographic provinces (CONABIO [15]). ALCH = Altos de Chiapas, ALT = Altiplano

Norte (Chihuahuense), ALS = Altiplano Sur (Zacatecano-Potosino), BAL = Cuenca del Balsas, BCA = Baja California, CABO = Del Cabo, CAL = California, CGM = Costa del Golfo, CPA = Costa del Pacífico, EVT = Eje Volcánico Transversal, OAX = Oaxaca, PET = Petén, SMOC = Sierra Madre Occidental, SMOR = Sierra Madre Oriental, SMS = Sierra Madre del Sur, SOC = Soconusco, SON = Sonora, TAM = Tamaulipas, YUC = Yucatán. Grid squares that comprise the Mexican Transition Zone are shown in gray.

## Materials and methods

The worldwide geographic distribution of 2703 genera of native flowering plants occurring in Mexico [8] was determined based on literature and on-line resources, mostly Tropicos (http://www.tropicos.org/) and The Plant List (http://www.theplantlist.org/). Because of their distribution these genera may be sorted out in four main floristic elements (Nearctic, Neotropical, Pan-American, and Widespread; Table 2). The set of Nearctic and Neotropical genera was selected because the distribution of these taxa has Mexico as their geographical boundary. Consequently, the area of greatest overlap of both elements would provide precise information about the transition zone.

Mexico was divided into 253 squares of 1° latitude and longitude (N = 253); however, the number was reduced by merging the smaller squares (those with less than half the surface of a total size square) with their neighboring squares so that the real number of squares analyzed was 175 (Fig 2). The number of genera of Nearctic and Neotropical affinity was determined for each of 175 grid squares. To define whether a square could be considered as part of the Nearctic or the Neotropical kingdom, a ratio of 3:1 was considered. That is, if a grid square recorded 3- times more Nearctic genera than Neotropical ones (or vice versa), it was assigned as part of that kingdom. When the ratio was less than 3:1, the square was considered as part of the Mexican Transition Zone.

Once the location of each grid square in one of the three regions (Nearctic, Neotropical, or MTZ) was established, its position was compared against the biogeographic provinces of CONABIO [15]. The use of the latter classification was preferred because it represents the consensus of several previous proposals and resulted from the discussion of about 20 experts who defined that biogeographic scheme.

The number of genera by geographical affinity was determined for each province. However, at this stage the number of genera was filtered, considering only those genera distinctive of the flora of Mexico (S1 Data). A genus was regarded as distinctive when half or more of its species were known from Mexico; for example, the genus *Calea* includes about 152 species worldwide, but in Mexico only 9 species are known, thus being a non-distinctive genus, whereas *Bursera* with 101 species in Mexico out of 120 species in total, is considered as a distinctive genus. With these selected genera and using the floristic provinces as Operational Geographical Units, an analysis of floristic similarity, using the Similarity Coefficient of Sorensen-Dice and UPGMA as grouping method, and a Parsimony Analysis of Endemicity (PAE) to identify possible areas of endemism were carried out. Within the three regions (Nearctic, Neotropical, and MTZ) support was sought for their biogeographic provinces as areas of endemism or biogeographic units defined by distinctive genera.

## Results

Twenty-one flowering plant families support the existence of a transition zone in Mexico's political territory; 10 of them characterizing the Nearctic Kingdom and 11 the Neotropical Kingdom (Table 1 and Fig 1). In addition to the two endemic families (Iteaceae and Setchellanthaceae), the other 19 have their northern and southern distribution limit somewhere in Mexico (Fig 1).

Among 2703 genera analyzed, 704 are considered to characterize the Nearctic Kingdom, while 772 the Neotropical Kingdom (Table 2). Among the genera defining the Nearctic Kingdom 232 genera are endemic to Mexico. In addition to these two elements, 323 genera registered in Mexico with a wider distribution in the Americas (Pan-American Element) are distributed further north, especially in the United States and south to South America; likewise, 904 genera with an extended range beyond the American continent, are considered here as the Widespread Element.

## The Mexican Transition Zone

The division of Mexico into 175 grid squares and the identification within each of the three elements analyzed (Nearctic, Neotropical, MTZ) showed that 78 belong to the Nearctic Kingdom, 51 to the Neotropical Kingdom, and 46 constitute the MTZ (Fig 2A). Fig 2B shows the biogeographic provinces located in both kingdoms and in the MTZ. Ten out of 19 provinces contain part of their surface in one of the two kingdoms and in the transition zone, suggesting that there are areas where equivalent proportions of both Nearctic and Neotropical elements intermingle in their territory.

The 46 grid squares that constitute the MTZ are located from the NE and NW of Mexico (24–24˚N and 28–29˚N respectively) to the south-central part (17–18˚N). In northeastern Mexico, MTZ includes part of the CGM, SMOR, and TAM biogeographical provinces, while in the northwest it includes parts of the provinces CPA, SMOC, and SON. In southern Mexico MTZ includes the south-central portion of the ALTS as well as parts of the provinces BAL, CPA, OAX, and SMS (Fig 2B).

Table 2 shows the distribution of distinctive genera among four floristic elements (with half or more of its species registered for Mexico). While the total proportion of Nearctic genera is somewhat lower than Neotropical genera, when distinctive genera are selected, the proportion is reversed so that the number of Nearctic genera becomes higher (575) than that of Neotropical affinity (282). Most Nearctic genera in Mexico have more species than the Neotropical ones. These 1152 distinctive genera were used in subsequent analyzes (S1 Data).

Table 3 shows the distribution of the distinctive genera among the 19 biogeographic provinces of Mexico. The number of endemic genera for each province is also indicated. Table 3 also shows that there are provinces where genera distributed in a single kingdom predominate, particularly in the Nearctic Kingdom. However, most provinces of the Neotropical Kingdom and the MTZ show high proportions of genera of both Nearctic and Neotropical affinity, without a dominance of one of them as noticeable as in the Nearctic Kingdom, except in the Yucatan Peninsula (PETE and YUC) provinces. Except for CAL which does not register any Mexican endemic genus, all other provinces contain from 4 (TAM) or 6 (YUC) to 82 (OAX) endemic genera (Table 3).

Among the 19 biogeographic provinces, 14 register at least one endemic genus within their territory; 10 of them are defined as areas of endemism because they include two or more endemic genera restricted to its territory (Table 4). In contrast, five provinces do not include any endemic genus. The biogeographic province with the highest number of endemic genera is ALTS (7), followed by SMS (5) and SMOC (4). CAL province, with no endemic genera, contains seven genera whose distribution in Mexico is only recorded from its territory, but extending their range north to the United States. Fig 3A illustrates the distribution of some of these genera according to their collecting localities linked by means of a minimum spanning network to illustrate their biogeographic track.

Several biogeographic provinces share at least one endemic genus between their territories (Table 5). Some pairs of provinces are grouped as areas of endemism even if they are not

**Table 3. Mexican biogeographical provinces (CONABIO [15]) and the number of distinctive genera assigned to the floristic kingdoms.** Total = Total genera recorded in the province.

| Kingdom | Province | Nearctic | Neotropical | Total |
|---|---|---|---|---|
| Nearctic or North American | Altiplano Norte (Chihuahuense) | 240 | 39 | 491 |
| | Altiplano Sur (Zacatecano-Potosino) | 373 | 95 | 598 |
| | Baja California | 160 | 11 | 286 |
| | Del Cabo | 93 | 12 | 220 |
| | California | 114 | 0 | 192 |
| | Sonora | 186 | 13 | 367 |
| | Tamaulipas | 77 | 16 | 235 |
| Neotropical or South American | Altos de Chiapas | 127 | 186 | 494 |
| | Costa del Golfo | 152 | 209 | 558 |
| | Costa del Pacífico | 243 | 199 | 604 |
| | Cuenca del Balsas | 197 | 151 | 497 |
| | Petén | 41 | 113 | 308 |
| | Soconusco | 151 | 233 | 563 |
| | Yucatán | 30 | 73 | 246 |
| Mexican Transition Zone | Eje Volcánico | 229 | 150 | 539 |
| | Oaxaca | 280 | 201 | 645 |
| | Sierra Madre Occidental | 231 | 104 | 500 |
| | Sierra Madre Oriental | 240 | 142 | 558 |
| | Sierra Madre del Sur | 220 | 157 | 506 |

located within the same biogeographic realm. Such is the case, for example, of ALTN and SMOC, which share the distribution of four genera, or of CPA and SMS that share two genera.

**Table 4. Endemic or distinctive genera with restricted distribution to one biogeographical province of Mexico (CONABIO [15]).** The genera whose collecting records were used to generate the biogeographic tracks shown in Fig 3A are indicated with an asterisk.

| Kingdom | Provincie | Genera |
|---|---|---|
| Nearctic or North American | Altiplano Norte (Chihuahuense) | *Fryxellia*\*, *Marshalljohnstonia*\*, *Raphanorhyncha*\* |
| | Altiplano Sur (Zacatecano-Potosino) | *Geissolepis*\*, *Planodes*\*, *Sanrobertia*\*, *Schaffnerella*\*, *Sohnsia*\*, *Stephanodoria*\*, *Strombocactus*\* |
| | Baja California | *Burroughsia*\* |
| | Del Cabo | *Carterella*\*, *Faxonia*\* |
| | California | *Achyrachaena*, *Anisocoma*\*, *Centrostegia*\*, *Githopsis*\*, *Pickeringia*\*, *Turricula*\*, *Umbellularia* |
| | Sonora | --- |
| | Tamaulipas | --- |
| Neotropical or South American | Altos de Chiapas | --- |
| | Costa del Golfo | *Byrnesia*, *Tuxtla*\* |
| | Costa del Pacífico | *Mexianthus*\*, *Tehuana*\* |
| | Cuenca del Balsas | X *Pachebergia* |
| | Petén | --- |
| | Soconusco | *Neomortonia*\*, *Vulcanoa*\* |
| | Yucatán | *Plagiolophus*\* |
| Mexican Transition Zone | Eje Volcánico | *Selloa*\* |
| | Oaxaca | --- |
| | Sierra Madre Occidental | *Brachystigma*\*, *Lasiarrhenum*, *Tacitus*\*, *Trichocoryne*\* |
| | Sierra Madre Oriental | *Epifagus*\*, *Velascoa*\* |
| | Sierra Madre del Sur | *Amoana*\*, *Dahliaphyllum*\*, *Glockeria*, *Lexarzanthe*\*, *Omiltemia*\* |

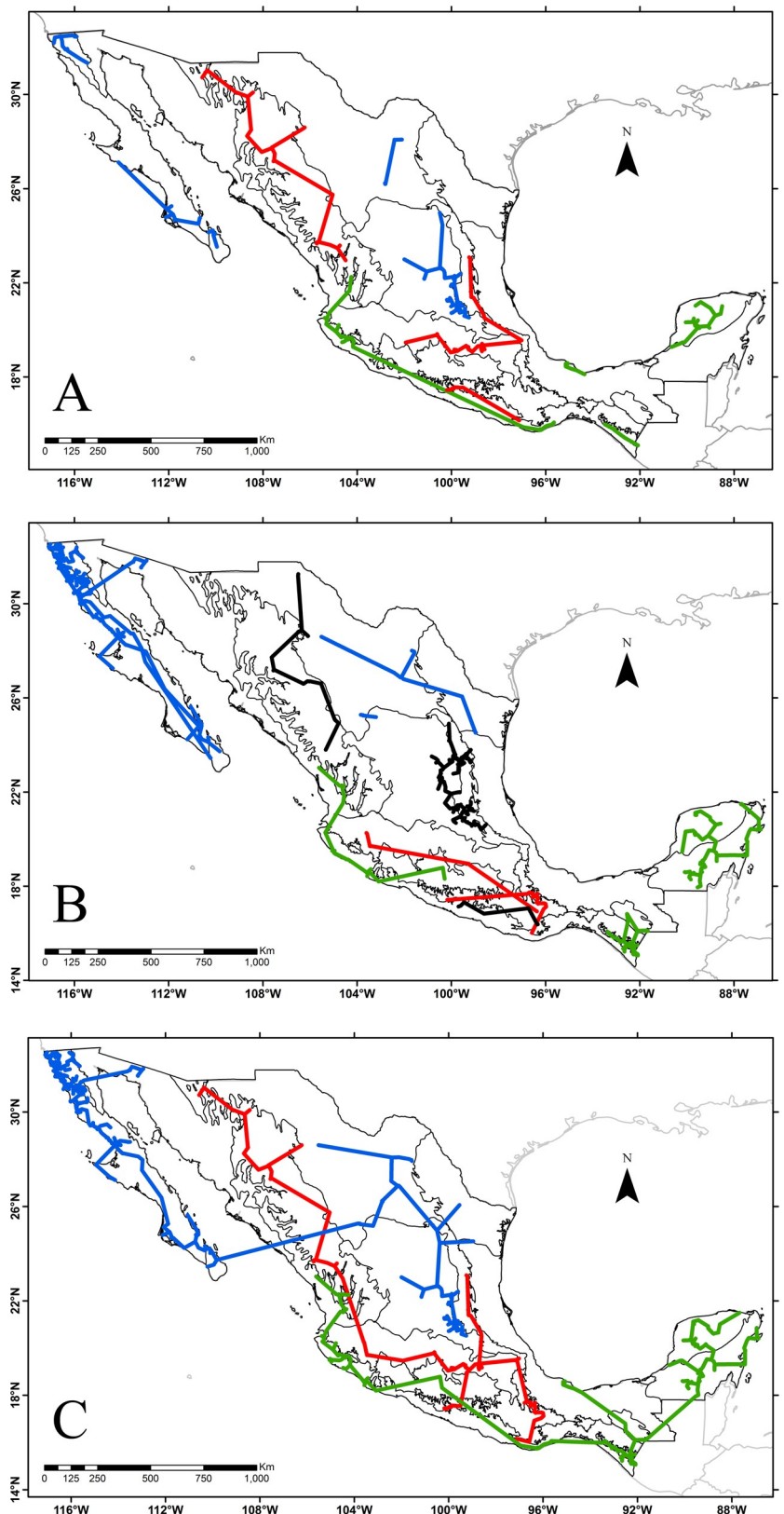

**Fig 3.** A. Endemic or distinctive genera of Mexico restricted to one biogeographic province of Mexico. B. Endemic or distinctive genera of Mexico with restricted distribution to two biogeographic provinces of Mexico. C. Addition of

distributions of all genera of the provinces considered part of the same kingdom or the Mexican Transition Zone. Biogeographic tracks of A and B combine the known distribution of the genera marked with an asterisk in Tables 4 and 5.

Table 5 also indicates the endemic or distinctive genera restricted to two contiguous biogeographic provinces, several of them forming areas of endemism for sharing more than two genera. Nine pairs of provinces register endemic genera in both territories, although one of them is a member of a floristic kingdom and the other province is part of the MTZ. For example, the ALTS (Nearctic Kingdom) and SMOR (MTZ) provinces share 6 genera, or the ALTN (Nearctic Kingdom) and SMOC (ZTM) provinces share 4 genera. Fig 3B illustrates, for some of these genera, their distribution according to their collecting localities linked by means of a minimum spanning network to illustrate their biogeographic track.

Fig 3C illustrates as a single track, the union of all the collecting sites of the distinctive or endemic genera restricted to one or two biogeographic provinces inside each region: Nearctic (blue), Neotropical (green), MTZ (red). Fig 3 shows how there are sets of endemic or distinctive genera that together support the boundaries between the Nearctic and Neotropical kingdoms; the MTZ is also identified by the generic proportions used to elaborate Fig 2A.

Fig 4 shows the groupings obtained with both clustering method and PAE. The cluster reveals the existence of four main groups (Fig 4A); one formed by the Northern Highlands

**Table 5. Endemic or distinctive genera with restricted distribution to two biogeographical provinces of Mexico (CONABIO [15]).** Provinces that are not located within the kingdom are highlighted in bold. Genera whose collecting records were used to generate the biogeographic tracks shown in Fig 3B are indicated by an asterisk.

| Kingdom | Provinces | Genera |
|---|---|---|
| Nearctic or North American | ALTN-TAM | *Limnodea\*, Pseudoclappia\*, Shinnersia\*, Villarrealia\** |
| | ALTN-ALTS | *Henricksonia\** |
| | ALTN-**SMOC** | *Batopilasia\*, Bealia\*, Cycloloma\*, Megacorax\** |
| | ALTS-**SMOR** | *Hoverdenia\*, Obregonia\*, Orthosphenia\*, Pseudonemacladus\*, Rzedowskia\*, Syringantha\** |
| | BCA-CABO | *Cochemiea, Coulterella\** |
| | BCA-CAL | *Bloomeria\*, Centromadia\*, Cneoridium\*, Corethrogyne\*, Chamaebatia\*, Dendromecon\*, Fremontodendron\*, Hoita\*, Jaumea\*, Ornithostaphylos\*, Romneya\*, Sarcodes\*, Stebbinsoseris\*, Stylomecon\*, Tetrapteron\*, Venegasia* |
| | BCA-SONO | *Asarina\*, Pelucha\*, Xylonagra\** |
| | CAL-SONO | *Ancistrocarphus, Atrichoseris, Prenanthella* |
| | SONO-**SMOC** | *Psoralidium* |
| | TAM-**CGM** | *Limnodea* |
| Neotropical or South American | ALCH-SOCO | *Plowmania\*, Rojasianthe\*, Squamopappus\*, Stanmarkia\*, Thornea\** |
| | BAL-CPA | *Apatzingania, Beiselia\*, Cymophora, Eryngiophyllum\*, Mexacanthus\*, Sciadodendron* |
| | BAL-**EVT** | *Cremnophila* |
| | CGM-CPA | *Guinetia* |
| | CGM-**OAXA** | *Barbieria* |
| | CGM-PETE | *Iseia* |
| | CGM-**SMS** | *Habroneuron* |
| | CPA-**EVT** | *Dirhamphis* |
| | CPA-SOCO | *Amolinia\*, Cymbosema\*, Christianella\*, Eizia\*, Plocaniophyllum\** |
| | CPA-**SMS** | *Anotea\*, Ortegocactus\** |
| | PETE-YUCA | *Attilaea\*, Canella\*, Oldenlandiopsis\*, Plagiolophus\*, Rachicallis\*, Strumpfia* |
| | SOCO-**SMS** | *Helleriella* |
| Mexican Transition Zone | EVT-SMS | *Buceragenia\*, Microepidendrum\** |
| | OAX-SMS | *Cyrilla\*, Hohenbergiopsis\*, Mocinnodaphne\*, Stramentopappus\** |

(ALTN) and Tamaulipas (TAM), another defined by the two provinces that constitute the Yucatan Peninsula, Petén (PETE) and Yucatán (YUC), and a third constituted by the three provinces circumscribed to the peninsula of Baja California, Baja California (BCA), Cabo (CABO), and California (CAL). The fourth group includes the remaining 12 provinces, 6 of them forming three additional groups, one that includes the Sierra Madre Occidental (SMOC) and Sonora (SONO), another that joins the provinces of the state of Chiapas (ALCH, SOCO), and the third joins the provinces Altiplano Sur (ALTS) and Sierra Madre Oriental (SMOR). The remaining six provinces are grouped with high similarity values, two of them characterizing the lowlands associated with the coastal plains of both slopes (CGM, CPA), another four including the mountainous regions of the south-central part of the country (EVT, OAX, SMS), which constitute the floristic province of the Meridional Mountain Ranges as defined by Rzedowski [11]. The last province of this group (BAL) is separated from the two previous groups by combining in its territory tropical elements in its low zones, a flora more similar to that found in the coastal plains, as well as many elements typical of temperate regions characteristic of the adjacent mountain ranges.

The cladogram (Fig 4B) shows groupings similar to those in the dendrogram (Fig 4A), especially in the pairs of provinces mentioned in the previous paragraph and identified with the same color. The dendrogram identifies eight pairs of provinces with a higher level of similarity between them, five of which are also recovered in the cladogram as areas of endemism. The other three clusters identified in the dendrogram are nested in sequence occupying close positions but differentiating themselves by the number of species that define the nodes. For example, the [OAX, EVT, SMS] cluster is characterized by six endemic genera exclusive of these regions (Table 5), two are shared between EVT and SMS, and other four genera are shared between OAX and SMS.

## Discussion

Lack of complete and reliable information about the flora of Mexico has prevented its evaluation of relationships and floristic affinities. It is now possible to analyze the whole-generic flora

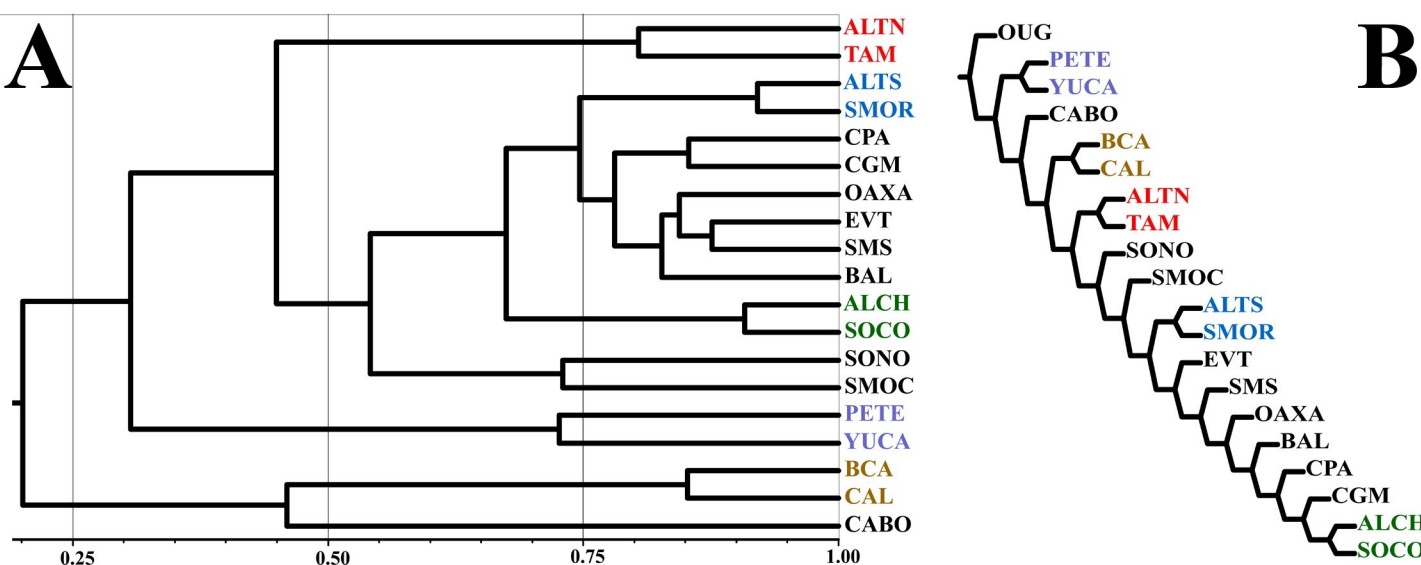

**Fig 4. Graphic representation of the relationships between biogeographic provinces and their generic flora.** A. Floristic similarities based on the Sorensen-Dice Similarity Coefficient and UPGMA as a grouping method (Software used: NTSYS-pc). B. Single cladogram obtained using Winclada (v. 1.99.08) with the heuristic option (Multiple TBR + TBR) using 10 replications. The same color is shown for matching provinces in both diagrams.

to identify the breadth of the Mexican Transition Zone and the biogeographic provinces included in the two floristic kingdoms that form the biogeographic boundaries in the country.

The endemic families of Mexico, as well as the endemic families of the Nearctic and Neotropical kingdoms with distribution in the country (Table 1), contain signatures to understand the evolutionary history of this region of the Americas. Many families originated at the end of the Cretaceous or in Tertiary times when much of Mexico constituted a peninsular end of Laurasia, isolated from Gondwana, and when some of the most important climatic changes happened worldwide. These events allowed the diversification of a significant number of taxa at the higher levels (families, genera). The identification of floristic cenocrons will undoubtedly provide a better idea of the evolution and diversification of the great floristic richness of the country. It will be important to separate the characteristic Neotropical groups from the lineages that represent tropical Laurasia at the end of the Cretaceous [4, 22] or those that migrated to North America after the formation of the Panama Isthmus.

Two biogeographic provinces located in the Nearctic Kingdom (ALTS and SONO) and three in the Neotropical Kingdom (BAL, CGM, and CPA) show grid squares whose surface contains equivalent proportions of genera with both Nearctic and Neotropical affinity (Fig 2B). Further studies will define how these grid squares could be defined based on important number of elements of both kingdoms. Some of them may behave as subtraction transition zones ('low overlap and progressive loss of taxa between regions'), while others behave as addition transition zones ('high overlap and progressive gain of taxa of each region'), as Ferro and Morrone [16] suggest.

The floristic similarity between provinces of the Mexican Highlands (ALTN and ALTS), coupled with the existence of endemic genera shared with the two important mountain ranges of northern Mexico (SMOC, SMOR) partly explain their association, although they are part of different biogeographical regions (Table 5 and Fig 2). According to Axelrod [23] many of the elements currently found in ALTN and ALTS are lineages diversified in mountain ranges, especially SMOC, which constituted the Madro-Tertiary geoflora.

Most genera shared between ALTS and SMOR are of Nearctic affinity (165 *vs* 76 of Neotropical affinity), although this overrepresentation is not clearly observed locally (grid squares used). In addition, among the 448 genera shared by both provinces (S1 Data), 327 of them are reported from temperate forests, tropical forests, and xerophytic scrublands, the three main biomes observed along the contact (transition) zone between these provinces. An additional analysis, considering only the distinctive genera occurring only in one or two biomes, shows that the correspondence between provinces is readjusted considering the environmental preferences of its members (Fig 5).

Biomes represent complex patterns of plant associations that result from interactions between climate and substrate [24] which can help connect the historical factors of plant distribution with environment, as discussed by Crisci et al. [25], to build the bridges between historical and ecological biogeography. The grouping of the biogeographic provinces by the exclusive genera restricted to one or two biomes (N = 253 genera) highlights the greater floristic similarity between ALTN and ALTS, although they share important percentages of genera with SMOC and SMOC.

A similar situation is observed between BAL and the surrounding mountain ranges (EVT and SMS). Floristically, BAL shares many genera with these two provinces, surely showing important interactions between their transition zones. However, Fig 5 shows how the distinctive and ecologically specific genera (1 or 2 biomes) in BAL correspond more to a tropical rather than to temperate affinity. Fig 5 shows that BAL is now associated with the representative provinces of the Neotropical Kingdom, which is also suggested by the cladogram of Fig 4B. These results agree more with the proposal of Espinosa et al. [17] to locate BAL within the

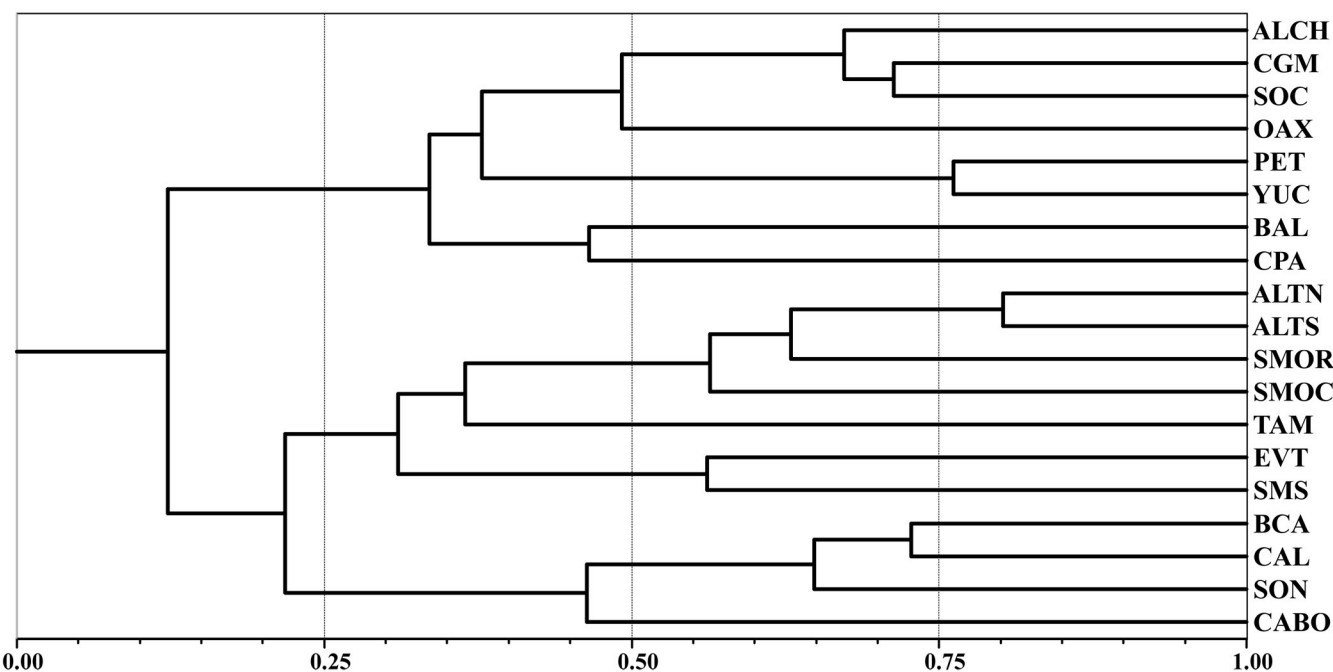

**Fig 5. Floristic similarities between the biogeographical provinces of Mexico considering only the distinctive genera (with half or more of their species in Mexico) registered in one or two biomes (N = 253).**

Neotropical Kingdom, an idea equally accepted now by Morrone's [14] (formerly considered by this author as part of the Mexican Transition Zone).

The Tamaulipas biogeographic province (TAM) constitutes the northern end of the coastal plain of the Gulf of Mexico in northeastern Mexico (Fig 2B). It is mainly characterized by a semi-arid environment, where there is a thorn-scrub considered of tropical affinity [11, 26], with many genera widely distributed in America or even in the Old World; however, the genera dominating the landscape in this province are of Nearctic affinity, such as *Castela*, *Eysenhardtia*, *Helietta*, *Leucophyllum*, *Nahuatlea*, *Neopringlea*, *Phaulothamnus*, *Porlieria*, or *Tiquilia*. Rzedowski [11] placed this province (called Northeast Coastal Plain) in its Mexican Xerophytic Region, within the Neotropical Kingdom. However, results indicate an overrepresentation of genera of Nearctic affinity (Fig 2A) linking this province more closely with ALTN (Fig 4) than with provinces from the Neotropical realm. In addition, these two provinces (ALTN and TAM) form an area of endemism, sharing four endemic genera (*Limnodea*, *Pseudoclappia*, *Shinnersia*, and *Villarrealia* (Table 5 and Fig 3B) that place them as part of the Nearctic Kingdom, as supported by Morrone [14].

## Boundaries between the Nearctic and Neotropical kingdoms

As indicated in Tables 3, 4 and 5, there are biogeographic provinces that define the floristic kingdoms and the Mexican Transition Zone. The Nearctic Kingdom is constituted by the provinces ALTN, ALTS, BCA, CABO, CAL, SONO and TAM, while the Neotropical Kingdom comprises the provinces ALCH, BAL, CGM, CPA, PETE, SOCO and YUC. In addition, the MTZ includes the mountain ranges from Oaxaca to northern Mexico (EVT, OAX, SMOC, SMOR, and SMS). Each floristic kingdom, in addition to their endemic families, records significant amounts of endemic genera to their territory, as shown in Tables 4 and 5. MTZ also registers regional endemic genera in its territory, highlighting its importance as a center of endemism.

There are important differences between the classifications of the provinces proposed here and the arrangement proposed by Rzedowski [11]. The provinces that constitute the Nearctic Kingdom (Table 3), comprise the North American Pacific and Mexican Xerophytic regions of Rzedowski [11], but the Tehuacán-Cuicatlán Valley (OAX in this work) is excluded, placing it in the MTZ, although genera of Neotropical affinity (Fig 5) predominate there. The ambivalent location of OAX is surely the result of combining dry tropical environments (such as those observed in the Tehuacán-Cuicatlán Valley) with temperate characteristic of the mountains of northern Oaxaca. Future studies in this biogeographic province will give a better appreciation of its floristic composition and define sub-regions within the province with greater precision, depending on the environments they contain.

The Caribbean Region of Rzedowski [11] is part of the Neotropical Kingdom (Table 3), except that in this work the mountains of Chiapas (ALCH and SOCO) are added to this kingdom which contain numerous elements typical of temperate mountain environments but of clearly Neotropical affinity. Finally, the Mesoamerican Mountain Region of Rzedowski [11] is practically the same as our MTZ, except for the mountains of Chiapas (province of the Transisthmic Mountains Range) that, as already indicated, are positioned here in the Neotropical Kingdom.

The results shown in Table 3 are in greater agreement with the proposals of Morrone [5, 14]. In particular, the Nearctic Kingdom corresponds faithfully with its Nearctic Region. The Neotropical Kingdom (Table 3) also contains the same provinces defined in the Neotropical Region by Morrone [14]; however, the same author [5] relocates his province of Chiapas (our CHIS) in the MTZ, an opinion not shared in this contribution. The MTZ as conceived by Morrone [14] is also recovered in this work.

## Is a unified classification of the biogeographic provinces of Mexico possible?

As discussed in this paper, there are apparently few discrepancies in the recognition of biogeographic units at the province level. The biogeographic provinces defined by CONABIO [15] constitute an excellent conceptual framework to discuss the biogeographical patterns of Mexican biodiversity. The differences in position in the biogeographic hierarchy of some provinces do not seem to cause major splits in the conception of biogeographic kingdoms or regions, which will allow a better circumscription and position in the future, as for instance, in the case of BAL, ALCH or SOCO. It would be equally important the definition of transition zones between the provinces, especially those that show a significant proportion of elements of both Nearctic and Neotropical affinity within their limits.

The unification of criteria about biogeographic units to be used must be discussed to achieve a better biogeographical scenario of Mexican biodiversity. For example, the province Meridional Mountain Ranges [11] is not equivalent and may not be compared with the biogeographic provinces EVT and SMS. Nor is it easy to reach conclusions by comparing units when some provinces are left out, for example, Morrone [5, 14] does not discuss the Tehuacán-Cuicatlán Valley province [11] or OAX, as proposed by CONABIO [15], and used in this work.

We agree with Parenti and Ebach's [27] idea that until an agreement is reached about the areas to be compared, little can be done to have a unified idea of the biogeographic history of Mexico. The occurrence of endemic families supports the recognition of two biogeographic kingdoms in this territory. Although there is no doubt that these two kingdoms are equivalent to the biogeographic regions used by zoologists, why it is not possible to unify the nomenclature? Similarly, if the nomenclature of the biogeographic provinces were unified,

communication and discussion would be more efficient and enriching when comparing biotic components of different taxonomic groups. The biogeographic provinces of CONABIO [15] have proved useful in various proposals that discuss biogeographic patterns for different taxa. Undoubtedly, for some provinces there are discrepancies that must be highlighted and evaluated with additional information. However, they can be a starting point to form a body of information to propose a geobiotic scenario [5] in the near future to prove that biota and earth make up a single biogeographic history.

## Supporting information

**S1 Data.**
(DOCX)

## Acknowledgments

Teresa Terrazas and two anonymous reviewers read the manuscript and made important contributions to improve it. Diego Juárez thanks CONACYT for a scholarship that made possible his work as a research assistant.

## Author Contributions

**Conceptualization:** Villaseñor José Luis.

**Data curation:** Villaseñor José Luis, Enrique Ortiz, Diego Juárez.

**Formal analysis:** Villaseñor José Luis.

**Funding acquisition:** Villaseñor José Luis.

**Investigation:** Villaseñor José Luis.

**Methodology:** Diego Juárez.

**Project administration:** Villaseñor José Luis.

**Software:** Enrique Ortiz.

**Supervision:** Villaseñor José Luis, Claudio Delgadillo-Moya.

**Validation:** Villaseñor José Luis.

**Visualization:** Enrique Ortiz, Diego Juárez.

**Writing – original draft:** Villaseñor José Luis, Claudio Delgadillo-Moya.

**Writing – review & editing:** Villaseñor José Luis, Claudio Delgadillo-Moya.

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
