## [Decision Letter · Decision Letter 0]

10 Mar 2020

PONE-D-20-02699

The breadth of the Mexican Transition Zone as defined by its flowering plant generic flora

PLOS ONE

Dear Dr. Jose Luis,

Thank you for submitting your manuscript to PLOS ONE. After careful consideration, we feel that it has merit but does not fully meet PLOS ONE’s publication criteria as it currently stands. Therefore, we invite you to submit a revised version of the manuscript that addresses the points raised during the review process.

Dear authors, reviewers were really positive on this ms evaluation. I send you their comments and suggest you to consider their comments specially from the anonymous reviewer.  

We would appreciate receiving your revised manuscript by Apr 24 2020 11:59PM. To enhance the reproducibility of your results, we recommend that if applicable you deposit your laboratory protocols in protocols.io, where a protocol can be assigned its own identifier (DOI) such that it can be cited independently in the future. For instructions see: http://journals.plos.org/plosone/s/submission-guidelines#loc-laboratory-protocols

We look forward to receiving your revised manuscript.

Kind regards,

Juliana Hipólito, Phd

Academic Editor

PLOS ONE

Journal Requirements:

"Work supported in part by the Universidad Nacional Autónoma de México - Programa de Apoyo a Proyectos de Investigación e Innovación Tecnológica (PAPIIT IN209519) to JLV.".

3.  We note that [Figures 1, 2, and 3] in your submission contain map images which may be copyrighted. All PLOS content is published under the Creative Commons Attribution License (CC BY 4.0), which means that the manuscript, images, and Supporting Information files will be freely available online, and any third party is permitted to access, download, copy, distribute, and use these materials in any way, even commercially, with proper attribution. For these reasons, we cannot publish previously copyrighted maps or satellite images created using proprietary data, such as Google software (Google Maps, Street View, and Earth). For more information, see our copyright guidelines: http://journals.plos.org/plosone/s/licenses-and-copyright.

1.     You may seek permission from the original copyright holder of Figures [1, 2, and 3] to publish the content specifically under the CC BY 4.0 license.  

Reviewers' comments:

Reviewer's Responses to Questions

**Comments to the Author**

1. Is the manuscript technically sound, and do the data support the conclusions?

Reviewer #1: Yes

Reviewer #2: Yes

2. Has the statistical analysis been performed appropriately and rigorously? 

Reviewer #1: Yes

Reviewer #2: Yes

3. Have the authors made all data underlying the findings in their manuscript fully available?

Reviewer #1: Yes

Reviewer #2: No

4. Is the manuscript presented in an intelligible fashion and written in standard English?

Reviewer #1: Yes

Reviewer #2: Yes

5. Review Comments to the Author

Reviewer #1: The ms represents an interesting contribution. I think that instead of Morrone (2005) the authors should cite the following:

Morrone, J. J. 2019. Regionalización biogeográfica y evolución biótica de México: Encrucijada de la biodiversidad del Nuevo Mundo. Rev. Mex. Biodivers. 90(e903980): 1-68.

This is relevant because in the latter the placement of the Balsas Basin province is more similar to the results obtained herein.

Reviewer #2: This is a detailed contribution on the biogeography of Mexico using information from plants. The authors emphasize that in comparison to proposals based on information from animals, those from plants are few. The analyses are appropriate for the study, and provide pertinent results. The text is essentially free of typos, although there are a few awkward phrases. The authors sufficiently support their conclusions about the characterization of the Mexican Transition Zone and affinities with the Nearctic and Neotropical regions of Mexico. However, I do have some concerns and recommend publication after the authors consider the following:

The authors should right at the start of the article define clearly and objectively their concepts of Nearctic and Neotropical. Most biologists have a generally understanding of these terms, but it needs to be stated how exactly they are being used in article because it has important implications for the rest of the paper.

The authors state: The number of genera of Nearctic and Neotropical affinity was determined for each of 175 grid squares. The authors need to elaborate on how this was determined and indicate the affinities of the genera that were included. This information is the basis of their paper and should also be provided as supplemental data.

Because I don’t completely understand the authors’ concept of Nearctic, I don’t understand their placement of many groups as being Nearctic. For example, Acanthaceae is an almost entirely tropical family. However, in the supplemental data many of its genera, such as Louteridium which is restricted to frost-free tropical areas, are classified as being Nearctic. I find this confusing. The same applies to Chiangodendron. This is a tree restricted to lowland tropical forests in the almost entirely tropical family Achariaceae. However, the authors classify it as Nearctic. This is counter intuitive. Furthermore, in the supplemental data the affinities (NEA, NEO, MTZ) of many genera are not given, e.g., Pectocarya, Sibara, Apteria. Was this information unintendedly left out or is it not possible to easily determine? The authors should double check that all information is accurate.

On page 9 the authors state “The territory of Mexico, for example, includes exclusively North American (Nearctic element) and South American (Neotropical element) as well as endemic families.” I think that they mean to say primarily instead of exclusively. If a family is exclusively South American, it by definition couldn’t occur in Mexico.

XPachebergia is a nothogenus based on a single individual that was determined to be a hybrid between Backebergia and Pachycereus. I don’t think that it should not be included in the analysis. It certainly is not comparable to other genera.

Instead of the category “cosmopolitan,” I suggest using the word widespread. Cosmopolitan has a connotation of occurring nearly throughout the entire world, which is the case for very few of the genera included in the category.

First sentence of abstract: “….have an identity provided by taxa….” Change to “…. are defined by taxa…..”

Fifth line of the abstract: “…agreement about the extent of this region.” Change to “...agreement about the limits and extent of this region.

Third paragraph of the introduction: “placing Mexico as one of such areas of endemism.” Change to “identifying Mexico as one such area of endemism.”

Page 12, Figure 2 caption. The authors state “Red: Neotropical Kingdom, Green: Mexican Transition Zone.” However, this is reversed and should be corrected. In the figure, red represents the Mexican Transition Zone and green the Neotropical Kingdom.

In various parts of the text: When starting a sentence, Figure should be written in full and not abbreviated.

Fourth line of the introduction: “These geological lineages….” Change to “These distinct geological histories….

Page 16, line 4: “...other 19 have their northern…” Change to “…the other 19 have their northern…”

Page 20, line 5: “that restrict their distribution to two” Change to “…restricted to two…”

Page 26, 1st sentence: “...constitutes the boreal end of the coastal...” Change to “…constitutes the northern end of the coastal…”

6. PLOS authors have the option to publish the peer review history of their article (what does this mean?). If published, this will include your full peer review and any attached files.

Reviewer #1: Yes: Juan J. Morrone

Reviewer #2: No

---

## [Author Response · Author response to Decision Letter 0]

29 May 2020

Juliana Hipólito, Ph. D.

Academic Editor

PLOS ONE

Dear Dr. Hipólito,

We have finished the revision of the manuscript PONE-D-20-02699 “The breadth of the Mexican Transition Zone as defined by its flowering plant generic flora”. The revised version has already been uploaded to the PLOS ONE platform.

 The corrections made to the manuscript are detailed below, starting with the editorial corrections and later with the changes made at request of the Reviewers. The changes indicated by reviewers are indicated in normal font and the modifications made to the manuscript are indicated in italics

Editorial corrections:

No changes were made to the statement of financial support. A paragraph was added in the cover letter with the financial support received for this investigation.

To enhance the reproducibility of your results, we recommend that if applicable you deposit your laboratory protocols in protocols.io, where a protocol can be assigned its own identifier (DOI) such that it can be cited independently in the future. For instructions see: http://journals.plos.org/plosone/s/submission-guidelines#loc-laboratory-protocols

Laboratory protocols were not applied, so there is no need to deposit them.

A rebuttal letter that responds to each point raised by the academic editor and reviewer(s). This letter should be uploaded as separate file and labeled 'Response to Reviewers'.

A separate file with the response letter to the editor was uploaded as 'Response to Reviewers'.

A marked-up copy of your manuscript that highlights changes made to the original version. This file should be uploaded as separate file and labeled 'Revised Manuscript with Track Changes'.

A file with the corrected manuscript was uploaded with the changes control active. The name of this file is 'Revised Manuscript with Track Changes'.

An unmarked version of your revised paper without tracked changes. This file should be uploaded as separate file and labeled 'Manuscript'.

The manuscript with updates without tracked changes was loaded into a file named 'Manuscript'.

If our paper is accepted, we have no objection to making the revision history of this manuscript public.

Style requirements were revised and the manuscript was structured accordingly. The format of the authors' affiliations was also corrected.

"Work supported in part by the Universidad Nacional Autónoma de México - Programa de Apoyo a Proyectos de Investigación e Innovación Tecnológica (PAPIIT IN209519) to JLV.".

The financing statement was corrected, which was as follows:

 “Work supported by the Universidad Nacional Autónoma de México - Programa de Apoyo a Proyectos de Investigación e Innovación Tecnológica (PAPIIT IN209519) to JLV. There was no additional external funding received for this study.”

The amended funding statement was included in the cover letter.

3. We note that [Figures 1, 2, and 3] in your submission contain map images which may be copyrighted. All PLOS content is published under the Creative Commons Attribution License (CC BY 4.0), which means that the manuscript, images, and Supporting Information files will be freely available online, and any third party is permitted to access, download, copy, distribute, and use these materials in any way, even commercially, with proper attribution. For these reasons, we cannot publish previously copyrighted maps or satellite images created using proprietary data, such as Google software (Google Maps, Street View, and Earth). For more information, see our copyright guidelines: http://journals.plos.org/plosone/s/licenses-and-copyright.

All figures were created by ourselves and no other map with prior copyright was used, so PLOS may publish it under Creative Commons Attribution License (CC BY 4.0). For this reason figures 1, 2 and 3 will not be removed from the shipment.

Figures were verified with the PACE tool; all of them, except figure 4, were successfully validated. PACE adjusted figure 4 with the following message: “Image file is flattened. TIF file is converted to a valid TIF file.”. The PACE Corrected figures were uploaded again to the PLOS ONE platform.

Reviewers' comments:

Reviewer #1:

The ms represents an interesting contribution. I think that instead of Morrone (2005) the authors should cite the following:

Morrone, J. J. 2019. Regionalización biogeográfica y evolución biótica de México: Encrucijada de la biodiversidad del Nuevo Mundo. Rev. Mex. Biodivers. 90(e903980): 1-68.

This is relevant because in the latter the placement of the Balsas Basin province is more similar to the results obtained herein.

According to Reviewer # 1, Morrone (2005) was changed to Morrone, JJ. Regionalización biogeográfica y evolución biótica de México: Encrucijada de la biodiversidad del Nuevo Mundo. Rev Mex Biodivers. 2019; 90: 1-68.

Reviewer #2:

To question 3: Have the authors made all data underlying the findings in their manuscript fully available? Reviewer # 2 answered "No".

In the supplementary material, the complete list of the genera used in this work is provided. It also includes the relationship of the biogeographic provinces of Mexico where they are found, as well as their biogeographic affinity. This list supports the results found.

This is a detailed contribution on the biogeography of Mexico using information from plants. The authors emphasize that in comparison to proposals based on information from animals, those from plants are few. The analyses are appropriate for the study, and provide pertinent results. The text is essentially free of typos, although there are a few awkward phrases. The authors sufficiently support their conclusions about the characterization of the Mexican Transition Zone and affinities with the Nearctic and Neotropical regions of Mexico. However, I do have some concerns and recommend publication after the authors consider the following.

The manuscript was revised again to improve its writing and understanding.

The authors should right at the start of the article define clearly and objectively their concepts of Nearctic and Neotropical. Most biologists have a generally understanding of these terms, but it needs to be stated how exactly they are being used in article because it has important implications for the rest of the paper.

The wording of some lines on pages 4 and 5 was changed to clearly and objectively define the concepts "Nearctic" and "Neotropical. The definitions were as follows:

“For the purposes of this study, Nearctic affinity elements (or North American sensu Cox [2]) are defined as the genera endemic to Mexico and those distributed from North America to Mexico and Central America. Furthermore, the Neotropical affinity elements (or South American sensu Cox [2]) comprises genera distributed from South America to Mexico. ”

The authors state: The number of genera of Nearctic and Neotropical affinity was determined for each of 175 grid squares. The authors need to elaborate on how this was determined and indicate the affinities of the genera that were included. This information is the basis of their paper and should also be provided as supplemental data.

The affinity of each genus was identified according to the definitions of the Nearctic and Neotropical elements given above. In the supplementary material provided, the affinity of each genera analyzed is specified.

Because I don’t completely understand the authors’ concept of Nearctic, I don’t understand their placement of many groups as being Nearctic. For example, Acanthaceae is an almost entirely tropical family. However, in the supplemental data many of its genera, such as Louteridium which is restricted to frost-free tropical areas, are classified as being Nearctic. I find this confusing. The same applies to Chiangodendron. This is a tree restricted to lowland tropical forests in the almost entirely tropical family Achariaceae. However, the authors classify it as Nearctic. This is counter intuitive. Furthermore, in the supplemental data the affinities (NEA, NEO, MTZ) of many genera are not given, e.g., Pectocarya, Sibara, Apteria. Was this information unintendedly left out or is it not possible to easily determine? The authors should double check that all information is accurate.

The wording on pages 4 and 5 was changed to improve the definitions of the Nearctic and Neotropical elements. These definitions were applied to each of the genera. In this way, although the Acanthaceae family in general terms is a tropical family, the 10 Louteridium species that occur in Mexico, are defined as Nearctic because they are endemic to Mexico or have a distribution from North America to Mexico and Central America. The same definition was used for Chiangodendron, an endemic genus of Mexico, and under the criteria established in this work, it is defined as of Nearctic affinity. A genus is Neotropical when its distribution is from South America to Mexico.

The information of the genera that by mistake did not show their biogeographical affinity was completed.

On page 9 the authors state “The territory of Mexico, for example, includes exclusively North American (Nearctic element) and South American (Neotropical element) as well as endemic families.” I think that they mean to say primarily instead of exclusively. If a family is exclusively South American, it by definition couldn’t occur in Mexico.

XPachebergia is a nothogenus based on a single individual that was determined to be a hybrid between Backebergia and Pachycereus. I don’t think that it should not be included in the analysis. It certainly is not comparable to other genera.

Following the suggestion, “exclusively” was changed to “primarily”.

Although XPachebergia is from hybrid origin (nothogenus), its taxonomic generic status was published and currently accepted in publications dealing with the Mexican flora. We prefer include it as one additional member of the Mexican generic flora.

Instead of the category “cosmopolitan,” I suggest using the word widespread. Cosmopolitan has a connotation of occurring nearly throughout the entire world, which is the case for very few of the genera included in the category.

“Cosmopolitan” was changed to “widespread”.

First sentence of abstract: “….have an identity provided by taxa….” Change to “…. are defined by taxa…..”

The change suggested by Reviewer # 2 was made.

Fifth line of the abstract: “…agreement about the extent of this region.” Change to “...agreement about the limits and extent of this region.

The change suggested by Reviewer # 2 was made.

Third paragraph of the introduction: “placing Mexico as one of such areas of endemism.” Change to “identifying Mexico as one such area of endemism.”

The change suggested by Reviewer # 2 was made.

Page 12, Figure 2 caption. The authors state “Red: Neotropical Kingdom, Green: Mexican Transition Zone.” However, this is reversed and should be corrected. In the figure, red represents the Mexican Transition Zone and green the Neotropical Kingdom.

The change suggested by Reviewer # 2 was made.

In various parts of the text: When starting a sentence, Figure should be written in full and not abbreviated.

The change suggested by Reviewer # 2 was made.

Fourth line of the introduction: “These geological lineages….” Change to “These distinct geological histories….

The change suggested by Reviewer # 2 was made.

Page 16, line 4: “...other 19 have their northern…” Change to “…the other 19 have their northern…”

The change suggested by Reviewer # 2 was made.

Page 20, line 5: “that restrict their distribution to two” Change to “…restricted to two…”

The change suggested by Reviewer # 2 was made.

Page 26, 1st sentence: “...constitutes the boreal end of the coastal...” Change to “…constitutes the northern end of the coastal…”

The change suggested by Reviewer # 2 was made.

---

## [Decision Letter · Decision Letter 1]

12 Jun 2020

The breadth of the Mexican Transition Zone as defined by its flowering plant generic flora

PONE-D-20-02699R1

Dear Dr. Jose Luis,

We’re pleased to inform you that your manuscript has been judged scientifically suitable for publication and will be formally accepted for publication once it meets all outstanding technical requirements.

Kind regards,

Juliana Hipólito, Phd

Academic Editor

PLOS ONE

Additional Editor Comments (optional):

Reviewers' comments:

Reviewer's Responses to Questions

**Comments to the Author**

1. If the authors have adequately addressed your comments raised in a previous round of review and you feel that this manuscript is now acceptable for publication, you may indicate that here to bypass the “Comments to the Author” section, enter your conflict of interest statement in the “Confidential to Editor” section, and submit your "Accept" recommendation.

Reviewer #1: All comments have been addressed

Reviewer #2: All comments have been addressed

2. Is the manuscript technically sound, and do the data support the conclusions?

Reviewer #1: Yes

Reviewer #2: Yes

3. Has the statistical analysis been performed appropriately and rigorously? 

Reviewer #1: N/A

Reviewer #2: N/A

4. Have the authors made all data underlying the findings in their manuscript fully available?

Reviewer #1: Yes

Reviewer #2: Yes

5. Is the manuscript presented in an intelligible fashion and written in standard English?

Reviewer #1: Yes

Reviewer #2: Yes

6. Review Comments to the Author

Reviewer #1: (No Response)

Reviewer #2: The authors have sufficiently addressed my concerns and made the necessary changes. They accepted most all of my suggestions, and when not, they justified their position (e.g., the continued listing of a nothogenus). The authors state that their definition of Neactic includes any element that is endemic to Mexico. This ignores taxonomic and biogeographical affinities, but if that is their definition, so be it, and it is clearly mentioned.

7. PLOS authors have the option to publish the peer review history of their article (what does this mean?). If published, this will include your full peer review and any attached files.

Reviewer #1: No

Reviewer #2: No

---

## [Editor Report · Acceptance letter]

16 Jun 2020

PONE-D-20-02699R1 

The breadth of the Mexican Transition Zone as defined by its flowering plant generic flora 

Dear Dr. Jose Luis:

I'm pleased to inform you that your manuscript has been deemed suitable for publication in PLOS ONE. Congratulations! Your manuscript is now with our production department. 

Kind regards, 

on behalf of

Dr. Juliana Hipólito 

Academic Editor

PLOS ONE